# Real-World Experience with Bezlotoxumab for Prevention of Recurrence of *Clostridioides difficile* Infection

**DOI:** 10.3390/jcm10010002

**Published:** 2020-12-22

**Authors:** Rosa Escudero-Sánchez, María Ruiz-Ruigómez, Jorge Fernández-Fradejas, Sergio García Fernández, María Olmedo Samperio, Angela Cano Yuste, Angela Valencia Alijo, Beatriz Díaz-Pollán, María Jesús Rodríguez Hernández, Esperanza Merino De Lucas, Oriol Martín Segarra, Carmen Sáez Bejar, Carlos Armiñanzas Castillo, Belén Gutiérrez-Gutiérrez, Dolors Rodríguez-Pardo, Antonio Ramos-Martínez, Julián Torre-Cisneros, Francisco López-Medrano, Javier Cobo Reinoso

**Affiliations:** 1Infectious disease Department, University Hospital Ramón y Cajal, 28034 Madrid, Spain; javier.cobo@salud.madrid.org; 2Spanish Network for Research in Infectious Diseases (REIPI RD16/0016/0011), Instituto de Salud Carlos III, 28029 Madrid, Spain; segarciafe@hotmail.com (S.G.F.); dolorodriguez@vhebron.net (D.R.-P.); julian.torre.sspa@juntadeandalucia.es (J.D.L.T.C.); 3Infectious Disease Unit, University Hospital 12 de Octubre, 28041 Madrid, Spain; rryruiz@gmail.com (M.R.-R.); flmedrano@yahoo.es (F.L.-M.); 4Pharmacy Department, University Hospital Ramón y Cajal, 28034 Madrid, Spain; jffradejas@salud.madrid.org; 5Microbiology Department, University Hospital Ramón y Cajal, 28034 Madrid, Spain; 6Infectious Disease Department, University Hospital Gregorio Marañón, 28009 Madrid, Spain; maria.olmedo.samperio@gmail.com; 7Infectious Disease Department, University Hospital Reina Sofía, Maimonides Institute for Biomedical Research (IMIBIC), University of Córdoba, 14004 Córdoba, Spain; cayuam@hotmail.com; 8Infectious Disease Department, University Hospital Puerta de Hierro, 28220 Majadahonda, Spain; angelavaal1@gmail.com (A.V.A.); aramos220@gmail.com (A.R.M.); 9Infectious Disease Department, University Hospital La Paz, 28046 Madrid, Spain; bdiazp14124@gmail.com; 10Infectious Disease Department, University Hospital Virgen del Rocío, 41013 Sevilla, Spain; rodriguezhernandezmariajesus@yahoo.es; 11Unit of Infectious Diseases, Alicante General University Hospital - Alicante Institute of Sanitary and Biomedical Research (ISABIAL), 03012 Alicante, Spain; merinoluc@gmail.com; 12Internal Medicine Department, University Hospital Fundación Alcorcón, 28922 Alcorcón, Spain; oriol.martinse@gmail.com; 13Infectious Disease Department, University Hospital La Princesa, 28008 Madrid, Spain; csaezbejar@yahoo.es; 14Infectious Disease Department, University Hospital Marqués de Valdecilla, 39008 Santander, Spain; carlos.arminanzas@scsalud.es; 15Unidad Clínica de Enfermedades Infecciosas, Microbiología y Medicina Preventiva, University Hospital Virgen Macarena, 41009 Sevilla, Spain; belengutiguti@hotmail.com; 16Departamento de Medicina, Universidad de Sevilla, 41009 Sevilla, Spain; 17Institute of Biomedicine Sevilla (IbiS), 41013 Sevilla, Spain; 18Infectious Disease Department, University Hospital Vall d’Hebron, 08035 Barcelona, Spain; 19Department of Medicine, School of Medicine, UCM. Instituto de Investigación Biomédica i+12, 28040 Madrid, Spain

**Keywords:** *Clostridium difficile*, *Clostridioides difficile*, *C. difficile* infection, bezlotoxumab, recurrence

## Abstract

Bezlotoxumab is marketed for the prevention of recurrent *Clostridioides difficile* infection (rCDI). Its high cost could be determining its prescription to a different population than that represented in clinical trials. The objective of the study was to verify the effectiveness and safety of bezlotoxumab in preventing rCDI and to investigate factors related to bezlotoxumab failure in the real world. A retrospective, multicentre cohort study of patients treated with bezlotoxumab in Spain was conducted. We compared the characteristics of cohort patients with those of patients treated with bezlotoxumab in the pivotal MODIFY trials. We assessed recurrence rates 12 weeks after completion of treatment against *C. difficile*, and we analysed the factors associated with bezlotoxumab failure. Ninety-one patients were included in the study. The cohort presented with more risk factors for rCDI than the patients included in the MODIFY trials. Thirteen (14.2%) developed rCDI at 12 weeks of follow-up, and rCDI rates were numerically higher in patients with two or more previous episodes (25%) than in those who had fewer than two previous episodes of *C. difficile* infection (CDI) (10.4%); *p* = 0.09. There were no adverse effects attributable to bezlotoxumab. Despite being used in a more compromised population than that represented in clinical trials, we confirm the effectiveness of bezlotoxumab for the prevention of rCDI.

## 1. Introduction

Recurrences remain the main challenge in the clinical management of patients with *Clostridioides difficile* infection (CDI) [1]. Although CDI frequency varies significantly across different cohorts and surveillance studies [2,3,4], clinical trials have consistently shown a recurrence rate of approximately 25% [5,6,7]. Compared with vancomycin, treatment with fidaxomicin has been associated with a significant reduction in recurrent CDI (rCDI) during the first 4 weeks after the end of treatment [5,6]. More recently, bezlotoxumab (a monoclonal antibody targeted against toxin B) has been commercialised. The MODIFY trials showed a 40% reduction in the rate of rCDI at 12 weeks of follow-up when bezlotoxumab was added to the standard-of-care antimicrobial therapy for CDI [7].

However, real-world evidence from publications on bezlotoxumab is extremely limited [8,9]. Real-world studies with bezlotoxumab are essential because the drug is probably being used in a population different from that represented in the MODIFY trials and under conditions of use that are also different from those in the clinical trials. Moreover, the rigid evaluation criteria of the clinical trials limit the applicability of the results to real-world settings. For instance, patients who did not achieve “cure” of a CDI episode at the end-of-treatment visit in a clinical trial are not evaluated as a recurrence event [5,6,7]. Meanwhile, in real life, these patients would receive longer treatments or may be considered cured as, although intestinal rhythm might not be completely normalised, they have clearly improved with treatment and may suffer an rCDI.

Real-world studies also provide important information on safety surveillance. In the MODIFY trials, a higher rate of heart failure was reported from patients treated with bezlotoxumab than from those treated with placebo [10]. Last, real-world studies might show subpopulations in which the drug is less effective [11]. In summary, real-world studies are necessary to help clinical decision-making, especially in the case of drugs with restricted access due to their elevated costs, as is the case for bezlotoxumab.

## 2. Materials and Methods

This retrospective, multicentre cohort study included all patients receiving bezlotoxumab infusion during the duration of antimicrobial treatment for CDI between July 2018 and July 2019 in 13 Spanish hospitals. The primary endpoint of the study was to describe the rate of rCDI during the 12 weeks after the end of antimicrobial treatment for CDI.

Medical records were reviewed by local investigators, and data were introduced in an online database. The study coordinator sent queries to local investigators to address all inconsistencies or presumed mistakes in the data. CDI episodes were classified according to the Infectious Diseases Society of America (IDSA) and Society for Healthcare Epidemiology of America (SHEA) criteria [12]. All patients receiving a haematopoietic progenitor transplant or a solid organ transplant, treatment with immunosuppressive agents, or chemotherapy or corticosteroids for more than 2 weeks or more than 20 mg per day for at least 1 week in the past 2 weeks, and patients with previous congenital or acquired humoural and/or cellular immunodeficiency were considered immunosuppressed. The severity of the episode was established according to the IDSA guidelines published in 2018 and to Zar et al. [12,13]. Recurrence was defined as a reappearance of the symptoms of the disease after symptom resolution from the previous episode, along with a positive test that demonstrated the presence of toxigenic *C. difficile* in the stool, during the follow-up [14]. Comorbidities and risk factors for recurrent CDI pre-established in the MODIFY studies were also recorded.

We compared the presence of five pre-established risk factors for rCDI from the MODIFY studies (age over 65 years, previous CDI episode, immunosuppression, infection due to a hypervirulent strain, and severe episode) in the current cohort with that in bezlotoxumab-treated patients from the MODIFY trial. We also included three other important variables for comparison: renal impairment (the most consistent comorbidity associated with rCDI) [15,16], positive direct toxin detection in faeces (also related to both recurrence rate and severity, in contrast to toxin-negative cases in which diagnosis is made by nucleic acid amplification tests (NAAT)) [17,18], and treatment with fidaxomicin (although it remains unknown if fidaxomicin plus bezlotoxumab is superior to vancomycin plus bezlotoxumab, fidaxomicin treatment itself is associated with a decrease in rates of CDI recurrence) [5].

Categorical variables are described through absolute and relative frequencies, while quantitative variables are described using the mean and standard deviation (SD) for those variables with normal distributions and medians and interquartile ranges (IQR) for those with non-normal distributions. To identify the risk factors associated with CDI recurrence after receiving bezlotoxumab during the 12-week follow-up period, the chi-squared test was used to analyse quantitative variables, while Student’s t-test and ANOVA were used for the analysis of a qualitative variable versus a quantitative variable according to the number of categories. The statistical significance for failure was defined as *p* < 0.05. The Kaplan–Meier method was used to describe the cumulative probability of rCDI stratified by CDI history before bezlotoxumab and analysed using the log-rank chi-squared test. All statistical analyses were performed using the Stata 13 statistics program. The investigation was carried out following the rules of the Declaration of Helsinki of 1975, revised in 2013. The study was approved by the Clinical Research Ethics Committee from the coordinating centre.

## 3. Results

Ninety-one consecutive patients from 13 centres were registered in the database. The median age of the patients was 71 years, 46 (50.5%) were men, and the median Charlson index was 4. Thirty-nine (42.9%) patients received bezlotoxumab during the first CDI episode, 28 (30.8%) during the first recurrence, and 24 (26.4%) during the second or later recurrences. Patients were classified according to current definitions [19] as healthcare facility-onset, healthcare facility-associated (HO-HCFA) in 39 (42.9%) patients, community-onset, healthcare facility-associated (CO-HCFA) in 35 (38.5%), community-associated (CA) in 11 (12.1%), and indeterminate in 6 (6.6%).

Most of the patients (72) were treated with vancomycin, and 32 of them received a tapered regimen. Table 1 shows the antibiotic treatments against *C. difficile*, treatment duration, and duration of the follow-up period starting at the end of the treatment against *C. difficile*.

As shown in Figure 1, the current cohort included patients with a higher risk of rCDI than those treated with bezlotoxumab in the MODIFY trials. They were older (67.0% vs. 49.9% aged over 65 years), had a higher proportion of previous CDI episodes (57.1% vs. 27.7%) and immunosuppression (61.5% vs. 22.8%), suffered more severe disease (44.9% vs. 15.6%), and more frequently experienced kidney failure (35.2% vs. 15.7%). Furthermore, the proportion of patients diagnosed by direct toxin detection was higher (72.5% vs. 49.0%) in the current cohort. In contrast, a lower number of patients was treated with fidaxomicin (3.8%) in the MODIFY trial than in the present cohort (14.3%). The rate of CDI produced by the hypervirulent strain was similar in both groups.

After a median follow-up time after the end of treatment of 74 (49–81) days and 84 (81–89) days after the infusion of bezlotoxumab, 13 out of 91 (14.3%) patients developed rCDI. The median time from the end of antibiotic therapy to recurrence was 19 (8–36) days. All recurrences occurred during the first 8 weeks.

Table 2 shows the variables in patients with and without recurrence. Although a statistically significant difference was not reached, the rate of rCDI was numerically higher in patients who had suffered two or more previous CDI episodes (25.0% vs. 10.4%; *p* = 0.09). Figure 2 shows the Kaplan–Meier curves for the time to rCDI analysis for patients who had suffered two or more previous episodes. Similarly, the recurrence rate was also higher in patients with the 027 ribotype (4 out of 10; 40%), although in our population, the strain ribotype was only determined in 48 patients.

We did not find any differences in the recurrence rates based on the anti-*C. difficile* drug received or whether standard or tapered–pulsed regimens had been used (Table 2). We also found no differences based on the microbiological technique used for the diagnosis, the age of the patients, or the severity of the episodes.

Thirteen patients (14.3%) had died by the time of the follow-up assessment point 12 weeks after the end of anti-*C. difficile* treatment. The median time to death after bezlotoxumab infusion and after the end of anti-*C. difficile* treatment was 28 (IQR 16–50) and 17 (IQR 2–46) days, respectively. Only in one patient was death directly related to CDI. Other causes of death were severe bacterial or fungal infections (five patients), progression of the underlying disease (two patients), progressive respiratory failure in a patient with COPD (one patient), massive haemoptysis (one patient), heart failure (one patient), and unknown (two patients). None of these 12 patients presented with rCDI before death. No adverse events apparently related to bezlotoxumab were reported by the investigators.

## 4. Discussion

Bezlotoxumab has been demonstrated to reduce the recurrence rate of CDI in clinical trials. However, there are only two published studies that have evaluated the effectiveness of bezlotoxumab in a real-world setting [8,9]. The present study confirms the effectiveness of bezlotoxumab in clinical practice. The observed rCDI rate (14.3%) was even lower than the rCDI rate reported in the pivotal clinical trial (17%). These results are remarkable considering the fact that bezlotoxumab is used (at least in Spain) in a much more compromised population than that represented in the MODIFY clinical trials. As shown in Figure 1, the patients included in our cohort presented a significantly higher risk of developing rCDI than those treated with bezlotoxumab in the MODIFY trials. The only factor that could have favoured a lower recurrence rate was the use of fidaxomicin for the treatment of CDI (somewhat more frequently observed in our cohort). However, only 13% of patients were treated with this drug; therefore, we cannot consider the influence of fidaxomicin in this study as a relevant factor. Additionally, the mortality during follow-up in our cohort (14%), higher than that described in the MODIFY trials (7%), indirectly shows a much more vulnerable population.

These results are comparable to those recently published from a North American cohort [9] and are apparently superior to those obtained in the only published European series [8]. The North American cohort (200 patients) consisted of nonhospitalised patients, with an age of 71 (median) similar to that of our cohort. However, it differs from the one presented here in that almost all patients (86.5%) had previous episodes of CDI and in a greater use of fidaxomicin as a treatment for the infection [9]. In contrast, the Finnish cohort (46 patients) comprised younger patients (mean age 66 years) that were frequently immunosuppressed and had numerous risk factors for recurrence (median, 4) [8].

Over one-third of our patients received tapered–pulsed antibiotic regimens against *C. difficile*. This approach is not standardised in clinical practice among participating centres. The most plausible explanation could be that the prescription of bezlotoxumab was typically (especially in the first months) dependent on approval by therapeutic committees or pharmacy services. In those cases, a prescription of longer regimens would ensure that the patient was still receiving treatment for *C. difficile* when approval was obtained. The use of tapered–pulsed regimens of vancomycin and fidaxomicin (not allowed in the MODIFY trials) could have influenced the results obtained in this study. However, we did not find any differences in the rate of rCDI when patients treated with conventional regimens (16.4%) were compared with those treated by the tapered or pulsed regimen (12.5%) (Table 2).

Since 13 patients died before the end of the 12-week follow-up, it could be said that the recurrence rate we presented could have been infraestimated. However, the median follow-up until death of these patients was approximately the same as the median time to rCDI in our cohort. Even if these patients were excluded from the analysis, the recurrence rate (16.7%) would be similar to the rate observed in the MODIFY trials.

The factors associated with bezlotoxumab failure in clinical practice should be investigated since they may help with using the drug in a more appropriate way. The results were numerically better in patients who had one or no previous episodes than in those who had two or more (Table 2). These results confirm those found by Hengel et al. [8] and suggest that bezlotoxumab should be used before multiple recurrence occurs, where otherwise the treatment of choice would be a faecal microbiota transplant. The rCDI rate was also numerically higher in patients associated with the 027 ribotype, but we only had this information for 48 patients, so we consider that the analysis of this variable is limited in our cohort. Due to the insufficient number of events, multivariate analysis could not be performed.

Bezlotoxumab has been demonstrated to be a safe drug. Heart failure was the cause of death for only one patient, and it occurred 12 weeks after the infusion of bezlotoxumab (day 86 postinfusion). Furthermore, none of the deaths occurred close in time to the infusion of the drug, and the researchers did not report any adverse events that seemed to be related to bezlotoxumab in their opinion.

Our study has certain limitations. This is a retrospective and multicentre cohort, which involves a risk of heterogeneity and loss of information. Since our definition of recurrence required microbiological confirmation of toxigenic *C. difficile*, some cases of rCDI might have been missed. However, all patients were managed by infectious diseases physicians skilled in the management and interpretation of the diagnostic tests for CDI.

## 5. Conclusions

Our study confirms the efficacy of bezlotoxumab for the prevention of rCDI in a real-world setting in Spain. The rCDI rate was comparable with that obtained in the MODIFY studies despite the presence of a much more compromised, at-risk population. The type of anti-*C. difficile* drug regimen did not influence the outcomes in our cohort. The results with bezlotoxumab were favourable regardless of age, severity, or comorbidities. However, the results appear to be worst in patients having two or more previous CDI episodes, which suggests that the use of bezlotoxumab should not be delayed.

## Figures and Tables

**Figure 1 jcm-10-00002-f001:**
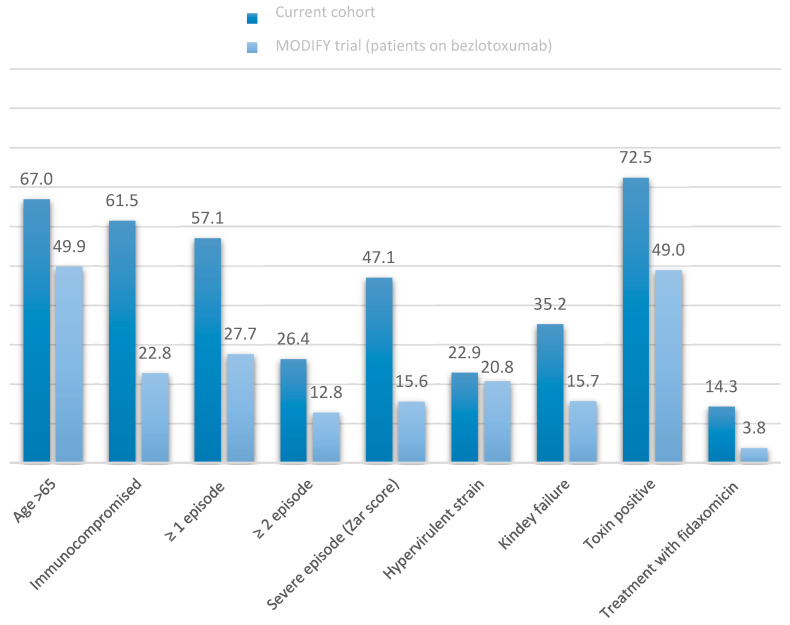
Comparison of variables in current cohort and bezlotoxumab-treated patients in MODIFY trial. Numbers show the percentage of patients.

**Figure 2 jcm-10-00002-f002:**
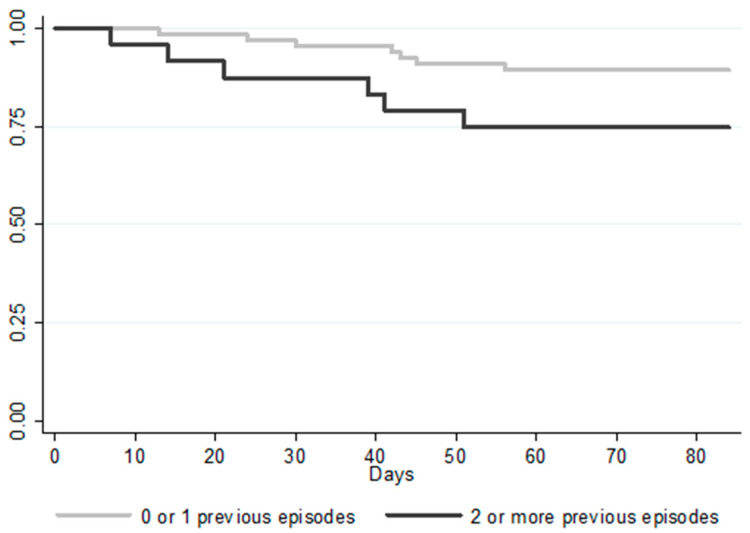
Kaplan–Meier plot of time to recurrent CDI.

**Table 1 jcm-10-00002-t001:** Duration of anti-*C. difficile* drug therapy.

Anti-*C. difficile* Treatment	Patients	Duration of Treatment	Follow-up after the End of Anti-*C. difficile* Treatment	Bezlotoxumab Infusion Time from the Start of Treatment
Vancomycin + metronidazole	5	10 (10–10)	76 (75–79)	2 (1–5)
Vancomycin	40	11 (10–14)	82 (77.5–86)	6.5 (3–10)
Vancomycin (tapered)	32	42 (35.5–55.5)	62 (45–73)	14 (3.5–29.5)
Fidaxomicin	9	11 (10–13)	79 (70.5–82)	5 (2–8)
Fidaxomicin (extend regimen)	4	24.5 (23–26.2)	79 (71–88)	12.5 (1.5–22)
FMT (after vancomycin)	1	9	79	12

FMT: faecal microbiota transplant. Time is indicated in days; median (Q1–Q3).

**Table 2 jcm-10-00002-t002:** Risk factors for recurrent *Clostridioides difficile* infection (CDI).

	Cohort	Recurrence	No Recurrence	*p*	95% CI
Number of patients	91	13	78		
Men	46 (50.5)	5 (38.5)	41 (52.6)	0.35	0.53–5.9
Age (years) *	71 (59–82)	68 (57–80)	72 (60–82)	0.96	0.96–1.04
Age > 65	61 (66.3)	8 (61.5)	53 (68.0)	0.65	0.22–2.54
Age > 85	17 (18.7)	3 (23.1)	14 (18.0)	0.66	0.33–5.64
Charlson index *	4 (2–6)	3 (2–5)	4 (2–6)	0.22	0.64–1.11
Kidney failure	32 (35.2)	4 (30.8)	28 (35.9)	0.72	0.22–2.81
Cancer	20 (22.0)	3 (23.1)	17 (21.8)	0.92	0.27–4.36
Leukaemia/Lymphoma	17 (18.7)	1 (7.7)	16 (20.5)	0.29	0.04–2.67
Any neoplasm	33 (36.3)	3 (23.1)	30 (38.5)	0.29	0.12–1.89
Liver disease	9 (9.9)	2 (15.4)	7 (9.0)	0.71	0.34–10.04
Intestinal inflammatory disease	6 (6.6)	1 (7.7)	5 (6.4)	0.86	0.13–11.34
Immunosuppression:	56 (61.5)	7 (53.9)	48 (62.8)	0.54	0.21–2.25
Chemotherapy	13 (14.3)	2 (15.4)	11 (14.1)	0.90	0.22–5.68
Steroids	14 (15.4)	1 (7.7)	13 (16.7)	0.42	0.05–3.49
Immunosuppressive drugs (not steroids)	16 (17.6)	1 (7.7)	15 (19.2)	0.33	0.04–2.91
Solid organ transplant	20 (22.0)	3 (23.1)	17 (21.8)	0.92	0.27–4.36
Previous CDI episodes:					
0	39 (42.9)	5 (38.5)	35 (44.9)	0.73	0.24–2.70
1	28 (30.8)	2 (15.4)	26 (33.3)	0.21	0.08–1.76
≥2	24 (26.4)	6 (46.2)	18 (23.1)	0.09	0.85–9.59
Proton pump inhibitor use	59 (64.8)	8 (61.5)	51 (65.4)	0.79	0.25–2.84
Previous antibiotic treatment	79 (86.8)	10 (76.9)	69 (88.5)	0.27	0.10–1.88
Classification of CDI episodes:					
CA	11 (12.1)	1 (7.7)	10 (12.8)	0.60	0.07–4.84
CO-HCFA	35 (38.5)	3 (23.1)	32 (41.0)	0.23	0.11–1.69
HO-HCFA	39 (42.9)	7 (53.9)	32 (41.0)	0.39	0.52–5.46
Indeterminate	6 (6.6)	2 (15.4)	4 (5.1)	0.19	0.55–20.59
Toxin positive	66 (72.5)	8 (61.5)	58 (74.4)	0.34	0.16–1.88
NAAT positive/toxin negative	25 (27.5)	5 (38.5))	20 (25.6)	0.34	0.16–1.88
IDSA severe or fulminant colitis	35 (38.5)	5 (38.5)	30 (38.5)	1.00	0.30–3.34
Severe (Zar)	41 (45.1)	7 (53.9)	34 (43.6)	0.49	0.46–4.91
Admitted to ICU	11 (12.1)	1 (7.7)	10 (12.8)	0.60	0.07–4.84
027 ribotype (based on 48 patients)	10 (20.8)	4 (44.4)	6 (15.4)	0.07	0.91–21.29
Concomitant antibiotics	25 (27.5)	1 (7.7)	24 (30.8)	0.12	0.02–1.52
Anti-*C. difficile* treatment:					
Vancomycin	40 (44.0)	5 (38.5)	35 (44.9)	0.67	0.23–2.56
Fidaxomicin	9 (9.9)	2 (15.4)	7 (9.0)	0.48	0.34–10.04
Vancomycin/metronidazole	5 (5.5)	1 (7.7)	4 (5.1)	0.71	0.16–14.99
Vancomycin (tapered)	32 (35.2)	4 (30.8)	28 (35.9)	0.72	0.22–2.81
Fidaxomicin extended–pulsed	4 (4.4)	0	4 (5.1)	0.40	-
Faecal microbiota transplant	1 (6.0)	1 (7.7)	0	0.01	-
Extended/pulsed–tapered treatments	36 (39.6)	4 (30.8)	32 (41.0)	0.49	0.18–2.26

* Values are indicated with median and interquartile range (Q1–Q3). CDI: *C.*
*difficile* infection; HO-HCFA: healthcare facility-onset, healthcare facility-associated; CO-HCFA: community-onset, healthcare facility-associated; CA: community-associated; NAAT: nucleic acid amplification tests; IDSA: Infectious Diseases Society of America; ICU: intensive care unit.

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
