# Peer review of "Real-World Experience with Bezlotoxumab for Prevention of Recurrence of Clostridioides difficile Infection"

_jcm, 2020, doi:10.3390/jcm10010002_

Round 1

Reviewer 1 Report

Introduction/Discussion: briefly summarize the results of the other 2 "real-world" studies and how these are different from this study, emphasizing new insights that these study may have.

Methods: From each site, does the decision to give bezlotoxumab based on clinician discretion? What parameters are used to favor prescribing vs not?  The study was approved by the Clinical Research Ethics Committee from the coordinating centre only; what was the rationale why this was not needed in the rest of the study sites?  Data coming from MODIFY trial- was these provided by the authors or culled from published literature?

Results/Discussion: Although it is informative to use the MODIFY trial as the comparator group, it would have been better to also include the local recurrence rates among similar groups of patients with CDI but did not get bezlotoxumab to confirm that the monoclonal antibody made a difference in the outcome of treatment in the target population.

Author Response

Introduction/Discussion: briefly summarize the results of the other 2 "real-world" studies and how these are different from this study, emphasizing new insights that these studies may have.
-.A new paragraph has been added in the discussion section

Methods: From each site, does the decision to give bezlotoxumab based on clinician discretion? What parameters are used to favor prescribing vs not?  The study was approved by the Clinical Research Ethics Committee from the coordinating centre only; what was the rationale why this was not needed in the rest of the study sites?  Data coming from MODIFY trial- was these provided by the authors or culled from published literature?
-.Yes, the treatment with bezlotoxumab was decided by each responsible physician according to the protocol of each centre. Therefore, there were no common criteria for the 13 participating hospitals.
-.As it is an observational retrospective study, the approval of the ethics committee of the study coordinating / promoting centre is required, to which the committees of the other participating centres adhere. Data from MODIFY trial was obtained from published articles.

Results/Discussion: Although it is informative to use the MODIFY trial as the comparator group, it would have been better to also include the local recurrence rates among similar groups of patients with CDI but did not get bezlotoxumab to confirm that the monoclonal antibody made a difference in the outcome of treatment in the target population
-.A new paragraph has been added in the discussion section

Reviewer 2 Report

Interesting paper.  Several things need clarification:

Were any Cdif patients excluded from the study.  If so, why?

bez. was used to treat several cases with recurrent Cdif.  (56?).  These included the 13 patients with previous bez. treatment?  These 13 would have been the only patients with multiple courses of bez.?

A summary table of results of bez. from MODIFY and references 8 & 9 and this publication would be interesting.

Author Response

Interesting paper.  Several things need clarification:

Were any Cdif patients excluded from the study.  If so, why?
-.All patients who received bezlotoxumab during the treatment of CDI were included in the study. Not all patients reached the 12 weeks of follow-up for evaluation of recurrence due to early death. However, this aspect is included in the discussion.

bez. was used to treat several cases with recurrent Cdif.  (56?).  These included the 13 patients with previous bez. treatment?  These 13 would have been the only patients with multiple courses of bez.?
-.Bezlotoxumab was used in 52 patients that had had previous CDI episodes but these episodes were not treated with bezlotoxumab. No patient received more than one course of bezlotoxumab.

A summary table of results of bez. from MODIFY and references 8 & 9 and this publication would be interesting.
-.It is not easy to collect the information from the three studies in a common table because the risk factors for recurrent CDI and the way of expressing the data of the population included are different in the three studies.